# 3-keto-DON, but Not 3-*epi*-DON, Retains the *in Planta* Toxicological Potential after the Enzymatic Biotransformation of Deoxynivalenol

**DOI:** 10.3390/ijms23137230

**Published:** 2022-06-29

**Authors:** Xiu-Zhen Li, Yousef I. Hassan, Dion Lepp, Yan Zhu, Ting Zhou

**Affiliations:** Guelph Research and Development Centre, Agriculture and Agri-Food Canada, 93 Stone Road West, Guelph, ON N1G 5C9, Canada; xiu-zhen.li@agr.gc.ca (X.-Z.L.); youhassan@yahoo.com (Y.I.H.); dion.lepp@agr.gc.ca (D.L.); yan.zhu@agr.gc.ca (Y.Z.)

**Keywords:** deoxynivalenol (DON), duckweeds, enzymatic biotransformation, stereoisomers, 3-keto-DON, mycotoxin, phytotoxicity, wheat

## Abstract

Deoxynivalenol (DON) is a secondary fungal metabolite that is associated with many adverse toxicological effects in agriculture as well as human/animal nutrition. Bioremediation efforts in recent years have led to the discovery of numerous bacterial isolates that can transform DON to less toxic derivatives. Both 3-keto-DON and 3-*epi*-DON were recently shown to exhibit reduced toxicity, compared to DON, when tested using different cell lines and mammalian models. In the current study, the toxicological assessment of 3-keto-DON and 3-*epi*-DON using *in planta* models surprisingly revealed that 3-keto-DON, but not 3-*epi*-DON, retained its toxicity to a large extent in both duckweeds (*Lemna minor* L.) and common wheat (*Triticum aestivum* L.) model systems. RNA-Seq analysis revealed that the exposure of *L. minor* to 3-keto-DON and DON resulted in substantial transcriptomic changes and similar gene expression profiles, whereas 3-*epi*-DON did not. These novel findings are pivotal for understanding the environmental burden of the above metabolites as well as informing the development of future transgenic plant applications. Collectively, they emphasize the fundamental need to assess both plant and animal models when evaluating metabolites/host interactions.

## 1. Introduction

There is a growing interest in biological detoxification approaches [1,2,3] as they constitute environmentally friendly methods to decrease the toxicological burden of many agricultural commodities inadvertently contaminated with toxins. It is anticipated that the prevalence and accumulation of agriculturally important toxins in cereals and grains will increase in the coming years as an unavoidable consequence of climate-change related effects [4,5]. This is particularly true for deoxynivalenol (DON), a *Fusarium* toxin produced by *F. culmorum* and *F. graminearum*, frequently encountered in corn, wheat, oats, barley, and rice, which has been on the rise in recent years [6,7,8].

The pursuit of biological systems that can be exploited for DON detoxification has intensified over the past four decades [1,9]. Earlier studies have engaged in seeking functional plant, fungi, bacteria, or bacterial consortia for this purpose and a number of enzymes/isolates with DON detoxification activities have been discovered and identified, including *Brachypodium distachyon* [10], *Eubacterium* sp. BBSH 797 [11], *Bacillus* sp. LS-100 [12], *Eggerthella* sp. DII-9 [13], *Slackia* sp. D-G6 [14], *Desulfitobacterium* sp. PGG-3-9 [15], and *Devosia mutans* 17-2-E-8 [16], which share the pathways of either UDP-glycosyltransferases, deepoxidation, or epimerization. Research efforts culminated in the recent discovery and characterization of multiple microbial DON catalytic systems, including DepA/DepB [17,18] and DdnA-Kdx-KdR [19]. Several metabolites of these systems have been reported as possible intermediate or final DON detoxification products including 3-keto-DON, 3-*epi*-DON, and DOM-1 [16,20,21,22,23]. For example, the DepA/DepB system oxidizes DON to 3-keto-DON, using pyrroloquinoline quinone (PQQ) as a cofactor, while a second reduction step occurs through a NADPH-dependent reaction, leading to the formation of 3-*epi*-DON [17,18].

In order to develop and optimize DON biodetoxification methods for industrial application, it is necessary to consider both the stability of DON under normal food/feed processing conditions [24], as well as the toxicological profiles of all resulting metabolites [17,18,25,26,27]. Several previous reports have described promising detoxification approaches [22,28,29,30] but were ultimately unable to deliver consistent industrial outcomes, due to the negative effects conferred by the catabolic by-products. For example, deepoxy-deoxynivalenol (DOM-1) has been shown to retain the same immune-modulatory effects of DON, while in the case of zearalenone and zearalenol, the by-product is in fact more toxic than the parental compound itself [31,32,33].

In the current report, we investigated the toxicity of 3-keto-DON and 3-*epi*-DON using two *in planta* model organisms, *Lemna minor* L., an aquatic freshwater plant commonly known as duckweed [34] and common wheat (*Triticum aestivum* L.). We also identified key plant metabolic pathways modulated by DON and 3-keto-DON, which suggest potential commonalties in their underlying molecular mechanisms. Shortly, in addition to their toxicological effects on animals, it is essential to also consider the role of DON and its metabolites in plant pathogenicity before any large-scale applications are sought [35,36].

## 2. Results

### 2.1. Lemna minor Studies

#### 2.1.1. The *Lemna minor* Bioassay Is a Valid and Sensitive System to Assess DON Toxicity

To assess the phytotoxic effects of DON and its two bacterial biodetoxification products, 3-keto-DON and 3-*epi*-DON, the aquatic macrophyte *L. minor* was used as an indicator-organism. Frond’s grown on Hoagland’s E+ medium containing each of the above compounds were investigated as described in the Materials and Methods section. Starting with six fronds per well as inoculum, the final number of fronds as well as their fresh-weights, were recorded as indicators of growth inhibition after 7 days. In order to assess the sensitivity of *L.*
*minor* to DON, a wide range of DON concentrations (0–5 µg/mL) was initially tested. After 7 days, the 0.5 µg/mL DON treatment caused a 23.2% decrease in total frond numbers, compared to the control group (18.88 ± 3.07 vs. 24.56 ± 4.45 fronds/well in controls), while the fresh-weight was reduced by 25% (13.75 ± 2.23 vs. 18.33 ± 4.38 mg/well in the control group) [Table 1(A)]. A dose-dependent response was observed, as 1.75 µg/mL DON resulted in a 60% and 66% decrease in total fronds and fresh-weights, respectively, whereas growth was completely inhibited at 5 µg/mL DON, with no increase of frond numbers.

#### 2.1.2. 3-keto-DON Exhibits *in Planta* Toxicological Effects against *L. minor* Comparable to DON

The phytotoxicity of 3-keto-DON was tested using the same concentrations range established for DON as reported above. 3-keto-DON caused an *in planta* growth inhibition comparable to, if not greater than, what was observed for DON in the *L. minor* bioassay. At a concentration of 0.5 µg/mL, 3-keto-DON caused a 51% reduction in total fronds (10.67 ± 1.7/well) compared to the negative control (21.75 ± 3.03/well), with a corresponding 50.7% decrease in fronds weight (7.19 ± 0.46 mg/well vs. 14.57 ± 2.74 mg/well in the control), as shown in Table 1(B).

Similarly, the inhibitory effects of 3-keto-DON followed a dose-dependent response, in which 1.75 µg/mL resulted in 70% reduction in frond numbers (with 66.8% decrease in fresh-weights), compared to the negative control, while the plant growth was completely inhibited by 2.5 µg/mL of 3-keto-DON. Furthermore, chlorosis of fronds was observed with plants grown in the presence of 3-keto-DON, the severity of which was correlating with 3-keto-DON concentration, as more bleached fronds were observed at higher 3-keto-DON concentrations (Figure 1). To verify that 3-keto-DON was not converting back to DON under the applied experimental conditions, samples were taken from wells treated with 5 µg/mL 3-keto-DON daily and were analyzed by HPLC. The results demonstrated that the 3-keto-DON concentrations did not change over the seven-day period of incubation (data not shown), indicating that the reversal of 3-keto-DON to DON by *L. minor* (or the associated microbiota) did not take place.

#### 2.1.3. High Concentrations of 3-*epi*-DON Do Not Negatively Affect *L. minor* Growth

Exposure to 3-*epi*-DON had no effect on *L. minor* growth, even when assayed at the highest concentration used for DON (5 µg/mL). Therefore, higher concentrations of 3-*epi*-DON were evaluated to determine its potential phytotoxic effects (Table 2). A concentration of 50 µg/mL induced a 41.6% reduction in frond numbers and 53.4% decrease of the fresh plants weight. The highest 3-*epi*-DON concentration tested (125 µg/mL) did not even completely inhibit the growth of *L. minor* (Figure 2A).

A regression analysis was carried out using an exponential model to examine the relationship between concentrations and growth inhibitions of DON and its metabolites. Due to the limited availability of 3-*epi*-DON, a correlation/regression design was used for the implemented toxicity-testing (using high concentrations) to add more statistical confidence to our observations. The resulting coefficient (A), constant (B), adjusted R square, and model significance values are listed in Table A1 while the regression curves are displayed in Figure 2A,B. Based on the presented exponential regression model, the *L. minor* IC_50_ values were determined to be 0.80 ± 0.14, 0.35 ± 0.12, and 34.04 ± 6.55 µg/mL for DON, 3-keto-DON, and 3-*epi*-DON, respectively, under our experimental conditions. The shown differences between IC_50_ means were found to be statistically significant (*p* < 0.05) based on the conducted independent-samples *t*-test.

### 2.2. Phytotoxic Effects of DON and Its Metabolites on Wheat Seedlings Are Similar to Those Observed with L. minor

The toxicity of DON and its biodetoxification products, 3-keto-DON and 3-*epi*-DON, on plants was further examined using a second *in planta* model consisting of germinated wheat-seeds [19]. The used bioassays are based on previous results that have been established in the literature showing the sensitivity and suitability of germinated wheat-seeds to test DON toxicity. Two winter varieties (AC Morley and AC Sampson) were utilized and concentrations of 10, 20, and 30 µg/mL were used for all three compounds [19]. As all three concentrations showed the same tendency, only the results of the 30 µg/mL are reported here.

Similar to *L. minor* bioassays, 3-*epi*-DON conferred no growth inhibition after 7 days incubation when compared to negative controls, whereas both DON or 3-keto-DON significantly reduced the fresh-weight and coleoptile length of wheat-seedlings (Figure 3 and Table 3). Root growth was also significantly affected by DON and 3-keto-DON but not 3-*epi*-DON.

The comparison of the two wheat varieties indicated that the *Fusarium*-susceptible variety, AC Sampson, was more sensitive towards the tested compounds than the moderately-resistant cultivar, AC Morley, as expected. Both DON and 3-keto-DON induced a pronounced decrease in AC Sampson wheat seedling coleoptile length by 31.0% and 33.2%, respectively, in comparison to the negative control (Table 3). The same was observed for the coleoptile fresh weights, which were 56.2% (DON) and 55.2% (3-keto-DON) of the negative control, as well as roots’ fresh weights, which were 17.5% (DON) and 15.5% (3-keto-DON) of the negative control. In the case of 3-*epi*-DON, all measured performance parameters of wheat seedlings were not significantly different from the negative controls (Table 3).

### 2.3. DON and 3-keto-DON, but Not 3-epi-DON, Significantly Modulated L. minor Transcriptome

To elucidate the underlying molecular mechanisms by which DON and its two bacterial metabolites exert their effects on *L. minor*, an RNA-Seq analysis was performed on plants exposed to DON, 3-keto-DON in addition to 3-*epi*-DON, and the results were compared to the negative control (media only).

Following quality filtering, the total high-quality reads per sample ranged from 66,376,776 to 100,496,994 with 87–96% of bases ≥ Q30 (Table A2). The proportion of reads that mapped to the *L. minor* draft genome ranged from 44.6–61.5%, of which 78.5–84.4% were mapped to predicted exon regions.

Gene expression analysis with HTSeq revealed a similar distribution of mean Fragments Per Kilobase of transcript per Million mapped reads (FPKM) between different treatment-groups (Figure 4A), indicating that the different treatments did not have a substantial effect on overall transcript abundance. Of the 22,382 predicted genes in the *L. minor* draft genome, a total of 14,262, 14,304, 13,746, and 14,254 were expressed (FPKM > 1) in the control, DON, 3-keto-DON, 3-*epi*-DON groups, respectively. The vast majority of these genes were co-expressed in all treatment groups (12,839) while 93, 236, 93, and 106 genes were uniquely expressed in the control, DON, 3-keto-DON, 3-*epi*-DON groups, respectively (Figure 4B).

The correlation of gene expression profiles within treatment groups was high (Pearson correlation coefficient R^2^ ≥ 0.93), with replicate samples exhibiting similar expression patterns, while average correlations between DON, 3-keto-DON, 3-*epi*-DON-treated samples, and the control samples were 0.863, 0.692, and 0.97, respectively. This indicated that expression profiles of control group samples were much more similar to the 3-*epi*-DON-exposed group than the DON and 3-keto-DON treatment groups. Furthermore, the previous observation was clearly supported by the clustering of treatment-groups, based on FPKM values, in which the control and 3-*epi*-DON groups clustered tightly together and separately from the two other groups (Figure 5A). This was also apparent in the multidimensional scaling (MDS) plot, in which Control and 3-*epi*-DON samples clustered closely, whereas DON and 3-*keto*-DON samples were separate (Figure 5B). Accordingly, very few differentially-expressed genes (DEGs; *p*-adjusted ≤ 0.05) were identified between the control and the 3-*epi*-DON groups, while a total of 786 and 3091 DEGs were significantly modulated by the DON and 3-keto-DON treatments, respectively. Of these, 23.6% were shared between the two groups (Figure 5C). Thus, our findings indicated that both DON and 3-keto-DON had profound and overlapping effects on the *L. minor* transcriptome, whereas exposure to 3-*epi*-DON had little to no impact.

The DEGs modulated by the DON and 3-keto-DON treatments were further annotated with Gene Ontology (GO) terms and Kyoto Encyclopedia of Genes and Genomes (KEGG) orthologies, and an over-representation analysis (ORA) was performed on these terms. A total of 8 and 20 KEGG pathways were enriched within the DON and 3-keto-DON modulated DEG sets, respectively, of which six were shared between the two treatment groups (Figure 5D).

The enriched KEGG pathways included a wide range of metabolic processes related to energy metabolism and photosynthesis, including starch and sucrose-metabolism, photosynthesis, photosynthesis–antenna proteins, pentose–phosphate pathway, carbon-metabolism, and carbon fixation in photosynthetic-organisms (Figure A1 and Figure A2).

Enrichment analysis of shared GO terms identified a similar set of metabolic functions affected by exposure to DON and 3-keto-DON. Specifically, exposure to DON resulted in the modulation of biological processes related mainly to photosynthesis and starch and sucrose metabolism. Similarly, a number of DEGs involved in photosynthesis-related processes and carbohydrate metabolism were enriched in the 3-keto-DON group, in addition to microtubule-based processes (Figure 5E).

Moreover, a number of enriched KEGG pathways included overlapping DEGs, as visualized by gene concept networks (Figure 6A,B and Figure A3), indicating that many of the DEGs may be involved in multiple biochemical pathways. In particular, a cluster of down-regulated DEGs in the 3-keto-DON group were associated with six related KEGG pathways, including carbon fixation in photosynthetic-organisms, glyoxylate- and dicarboxylate-metabolism, carbon-metabolism, amino acid (glycine, serine and threonine) metabolism and the pentose-phosphate pathway. Fourteen enriched KEGG pathways were associated with downregulated DEGs in the 3-keto-DON group, while five were associated with upregulated genes. This included the mitogen-activated protein kinase (MAPK) pathway, in which 19 DEGs were upregulated (Figure A2K). Interestingly, this was the only signaling pathway modulated by 3-keto-DON, which, unlike biochemical pathways, has the potential for inducing wide-ranging effects through global regulation of downstream effectors. Overall, these results were consistent with the GO term enrichment analysis, and together suggest that both DON and 3-keto-DON have wide-ranging effects on plant physiology, impacting both photosynthesis and cellular respiration, as well as the MAPK signalling pathway.

## 3. Discussion

A growing interest in the identification of microbial and enzymatic approaches to address the accidental contamination of agricultural commodities (food and feed) with mycotoxins has led to the isolation of multiple organisms and microbes that produce various oxidative and/or reductive mycotoxin metabolites. These metabolites generally exhibit altered physicochemical characteristics in comparison to the parental compounds, mostly with undefined toxicological profiles. It is necessary to rigorously evaluate the toxicity of these metabolites in order to develop safe and practical industrial applications. In many cases, a standardized toxicity assessment is performed using model-organisms, such as mammalian cell lines or rodent models, to highlight the reduced toxicity. However, the unique relationship between specific-hosts with each presented metabolite is seldom considered.

Through recent investigations of a bacterial enzymatic system that catalyzes the epimerization of DON [17,18,21], we identified two specific metabolites, namely 3-keto-DON and 3-*epi*-DON, of which 3-keto-DON is an intermediate while 3-*epi*-DON accumulates as the biotransformation’s end-product. Our earlier results highlighted the reduced toxicity of both compounds [20,26], although indicating that the complete conversion of DON to 3-*epi*-DON is more advantageous for achieving adequate detoxifications from the empirical point of view [26].

A previous report using mitogen-induced and mitogen-free proliferations of mouse spleen lymphocytes [9] suggested the sufficiency of the DON to 3-keto-DON step on its own [9,37], while BrdU bioassays using 3T3 cells indicated a substantial difference of the toxicity profiles between 3-keto-DON and 3-*epi*-DON with an estimated 45 fold further decrease in toxicity under the light of a complete epimerization of the C3 carbon [26].

The reduced toxicity of 3-*epi*-DON was confirmed by several recent studies. In intestinal explants, the intestinal lesions induced by DON treatment were not observed in explants treated with 3-*epi*-DON [38]. An animal trial using piglets demonstrated also that 3-*epi*-DON is not toxic for piglets, one of the most DON sensitive animals [39].

To settle the observed uncertainties in toxicological profiles of the above metabolites, address the magnitude of toxicity-reduction in the above-described epimerization pathway, and to finally elucidate the *in planta* toxicological potency of 3-keto-DON; we pursued this work using two plant model systems, duckweed and domesticated wheat.

Our results clearly show that 3-keto-DON does not only retain its capacity to act as a strong plant toxin but also has the ability to suppress stem growth and roots development in wheat seedlings as well as other plants. In contrast, the effects of 3-*epi*-DON on wheat seedling growth was negligible. In fact, wheat seedlings exhibited better growth in the 3-*epi*-DON treatment group than the negative control. The same was also observed in regard to the fresh weight of measured vegetative parts of 3-*epi*-DON treatment compared to DON treatment.

Mechanistic insights into the plant toxicity of DON and 3-keto-DON were also gained in the current study through the transcriptomic analysis of *L. minor* exposed to these compounds. Both DON and 3-keto-DON induced wide-ranging but similar transcriptomic changes within the *L. minor* model system, while exposure to 3-*epi*-DON had negligible effects on gene expression of essential genes. These results are consistent with the observed effects of these compounds on *L. minor* growth, suggesting that, in contrast to DON and 3-keto-DON, 3-*epi*-DON does not induce major physiological changes in *L. minor*. Interestingly, exposure of *L. minor* to 3-keto-DON resulted in a greater number of modulated genes than to DON itself, suggesting that the mode of action of 3-keto-DON may slightly differ from DON. The predicted functions conferred by the DEG profiles, as determined by over-representation analysis of GO and KEGG annotations, overlapped between DON and 3-keto-DON, affecting several systems including photosynthesis, DNA replication, and metabolic processes, although 3-keto-DON affected additional metabolic pathways as well as MAPK signaling, a pathway that was reported previously to be activated by DON [40].

While a number of previous transcriptomic studies have examined exposure to DON in animal model systems [41,42,43,44,45], a few *in planta* analyses have been conducted [46,47,48], and none have been performed on 3-*epi*-DON and 3-keto-DON. An RNA-Seq analysis of wheat (*Triticum aestivum* L.) in response to *Fusarium pseudograminearum* infection identified ~2700 differentially expressed genes, representing functions related to pathogenesis, host defense mechanisms, transcription factors, transporters and UDP glycosyltransferases [46]. Hofstad et al. [48] investigated the response of wheat spikelets to DON by RNA-Seq in both DON-resistant and DON-susceptible genotypes, in which they identified 1228 and 1012 DEGs, respectively. The 281 DEGs induced by DON in both genotypes again included genes related to detoxification and transport, including glutathione-S-transferases, UDP-glycosyltransferases, adenosine triphosphate-binding cassette (ABC) transporters, and cytochrome P450 genes [48] but minimal overlap with a transcriptomic analysis of DON-exposed barley was found. Similarly, the transcriptomic response of *L. minor* appears to differ from that of wheat and barley, in terms of the major biochemical pathways affected, despite the apparently similar physiological responses of *L. minor* and wheat to DON and its metabolites observed in the current study.

The role and ability of 3-keto-DON to induce MAPK pathway genes is consistent with previous studies that associated the adverse effects of DON with the induction of MAPK signaling pathway [49]. The toxicity of DON and its ability to induce histological changes within the intestine was particularly highlighted and connected for years with MAPK ERK 1/2, p38, and JNK activation [50].

The relationship between structure–activity of DON and its metabolites needs to be further studied. It was suggested that the reduced toxicity of 3-*epi*-DON is mainly due to weaker binding affinities to molecular targets (including ribosomes) in comparison to DON [51,52,53]. 3-keto-DON is also anticipated to have a weaker binding affinity to ribosomes relative to DON, highlighting the uncertainties of toxicity-models that are built solely upon binding-affinities [52,53].

The detected differences in wheat-seedling response of DON-susceptible (AC Sampson) and moderately resistant (AC Morley) varieties in this study are noteworthy. AC Morley has a native source of *Fusarium* head blight (FHB) resistance [54]; hence, the differences between the DON-treated seedlings and the control, while being significant, they did not reach the same severity level that was observed in the AC Sampson cultivar (Figure 3). Interestingly, 3-keto-DON affected AC Morley roots growth to a lesser degree too. As the FHB resistance is considered a quantitative genetic trait [55] and in the light of the above findings, it is reasonable to conclude that 3-keto-DON was negatively affecting more than one target gene(s)/pathway(s) that are involved in wheat-seedlings growth and roots development and/or FHB resistance.

Moreover, the current findings of this study highlight the potency of 3-keto-DON in plants. While transforming DON to 3-keto-DON might be acceptable for animal-feeding applications based on the earlier studies [9], such a transformation may not be useful to address DON associated-toxicities and adverse effects in any plant applications (transgenic and non-transgenic) as the DON to 3-keto-DON conversion will exert a detrimental influence on plant growth resulting possibly in an intensified field-pathogenicity in comparison to DON.

Collectively, these findings also indicate that the end-product of DON transformation by the *Devosia mutans* 17-2-E-8 isolate, 3-*epi*-DON, is safe and suitable for plant-based applications.

Altogether, the above findings highlight the importance of considering the targeted-hosts/organisms interactions when planning for actual applications, as some might yield better outcomes through utilization of the entire enzymatic pathway with 3-*epi*-DON as final product while others could be optimized with the DON to 3-keto-DON step only.

Furthermore, the source and availability of needed enzymatic-cofactors, such as PQQ and NADPH, are another essential element to consider when planning DON biotransformation applications at the industrial scale. The high cost(s) of these cofactors, especially NADPH, would place an empirical hurdle in the development of commercial applications unless regeneration systems are adopted. Moreover, the involved two-step reactions not only increase the work stream in practicality but also may introduce some uncertainties into the functional efficiency of the enzymes given the possibility of surrounding environmental constituents interfering in enzymes/cofactors performance. Part of these challenges can be addressed by applying novel strategies including protein engineering and enzyme immobilization.

## 4. Materials and Methods

### 4.1. Chemicals and Materials

Media and ingredients used for this study were purchased either from Fisher Scientific (Nepean, ON, Canada), Sigma-Aldrich (Oakville, ON, Canada) or Difco (Sparks, MD, USA). DON (Cat. #CD0228) and 3-keto-DON (Cat. #CK0013) were both obtained from Triple Bond (Guelph, ON, Canada) and stored in −20 °C freezer until usage. The studied 3-*epi*-DON was purified and its identity confirmed as previously reported [20]. Mycotoxin stocks (as well as standards intended for HPLC analysis) were dissolved in acetonitrile at 1 mg/mL and stored at −20 °C. Working HPLC solutions of DON, 3-keto-DON, and 3-*epi*-DON (10 µg/mL) were prepared in acetonitrile too and stored at 4 °C briefly before analysis.

DON and 3-*epi*-DON preparations used in toxicity-testing were dissolved in sterile water while 3-keto-DON was dissolved in dimethyl sulfoxide (DMSO), all at a concentration of 10 mg/mL and stored properly at −20 °C. In a later time and before usage, working solutions (1 mg/mL) were prepared freshly to be incorporated into the “*Lemna* medium”. Proper negative-controls (containing DMSO only) were utilized to account for DMSO toxicity (if any) and to facilitate direct simultaneous comparisons.

### 4.2. Duckweed (Lemna minor) Bioassays

The aquatic plant *Lemna minor* L. used in this study was obtained from the Canadian Phycological Culture Centre (Cat. #CPCC 490; Waterloo, ON, Canada). The plant was grown in foam plug-capped Erlenmeyer flasks (250 mL) containing 100 mL of Hoagland’s E+ medium as recommended by the culture provider. Flasks were maintained in a growth chamber at 22 ± 2 °C under a light regime of 16 h:8 h (light/dark) and illumination of 70 μmol·m^−2^∙s^−1^. Plants were sub-cultured every 10–12 days by transferring six plants to 100 mL of fresh growth medium. To obtain sufficient and homogenous plant starting-population, one week ahead of each experiment; a master pre-culture was initiated with six mature plants (3–4 fronds/plant) in a flask containing 100 mL of fresh medium.

For duckweed’s growth-inhibition bioassays, the bioassays were performed in sterile 24-well plates containing 2 mL of Hoagland’s E+ medium in each well. Mycotoxin stocks were added to the Hoagland’s E+ medium as well as proper blank-controls. DON and 3-*epi*-DON stock solutions were prepared in water as mentioned above while the 3-keto-DON stock solution was prepared in DMSO. Final test concentrations spanned 0, 0.5, 1, 1.75, 2.5 and 5 µg/mL for DON and 3-keto-DON and 0, 2.5, 5, 25, 50 and 125 µg/mL for 3-*epi*-DON in test wells. In all replicates, 10 μL of DMSO were introduced into each media-well to account for DMSO toxicity (if any). Six fronds of actively growing *L. minor* were placed in each well. The experiment was performed in quadruplicate over a period of 7 days, and the outcomes were analyzed based on duckweed frond-growth. The final number of fronds was counted as well as determining frond’s fresh weight. On the 7th day of growth, the media of each individual-well were sampled to a separate vial, filtered through a 0.45 µm PVDF filter (VWR International, Radnor, PA, USA), and analyzed by high performance liquid chromatography (HPLC). Likewise, the (0) time of each control (DON, 3-keto-DON, and 3-*epi*-DON) was also analyzed by HPLC as described below.

### 4.3. Common Wheat (Triticum aestivum) Bioassays

Common wheat (*Triticum aestivum* L.) seeds were obtained from the Department of Plant Agriculture, University of Guelph, Ridgetown Campus (Chatham-Kent, ON, Canada). The two wheat cultivars that were used: AC Morley (Winter wheat, moderately resistant to *Fusarium*) and AC Sampson (Winter wheat, susceptible to *Fusarium*). The phytotoxicity testing of DON, 3-keto-DON, and 3-*epi*-DON were conducted according to a seed-germination protocol adopted from Ito et al. [19] with minor modifications.

In brief, seeds were sterilized with 0.5% sodium hypochlorite and 0.3% Tween 20 in water for 3 min before washing thoroughly using sterile-water for 5 times, followed by one hour incubation in sterile-water. Subsequently, seeds were placed on paper-towels in closed Petri dishes and were incubated at 28 °C (in dark) for additional 24 h.

For gellan gum-soft gel media (GSGM) preparation, 200 µL of 15 mM CaCl_2_ containing the tested toxin/metabolite in addition to 3% DMSO were mixed with 400 µL of 0.3% gellan gum in 2-mL tubes, these mixtures were left for 24 h at room temperature to solidify before usage.

Germinated wheat-seeds were inoculated onto 600 µL of GSGM containing DON, 3-keto-DON, or 3-*epi*-DON. The germ of the seed was placed with face up while the lid of the hosting tube (2 mL) was removed. The entire assembly was then placed into a sterile glass tube (200-mm length and 18-mm internal diameter) and aseptically incubated at 28 °C in the dark. Seedlings (*n* = 15) were grown for 7 days on GSGM containing 30 µg/mL DON, 3-keto-DON, or 3-*epi*-DON and 1% DMSO before the final assessment.

On the assessment day, the seedlings were cut off the seeds and both fresh-weights and the length of the coleoptiles were measured individually for each seedling. Root-tissues were washed also, blot dried with paper towels, and the fresh-weight was determined. As mentioned above, fifteen seedlings were tested for each treatment in parallel to control seedlings (growing on GSGM containing 1% DMSO only). Experiments were repeated three times.

Finally, and in a separate experiment, we sampled a mixture of media (GSGM as well as Hoagland’s E+) containing 5 µg/mL of DON and metabolites (3-keto-DON and 3-*epi*-DON) during the entire duration of the experiment (either incubated at room temperature or at 28 °C) in order to check the stability of the three tested chemicals.

### 4.4. HPLC Analysis of DON, 3-keto-DON, and 3-epi-DON

Media samples were diluted 10 times using acetonitrile:water (2:8 by volume) and passed through 0.45 μm filters (Cat. #6765-1304; Whatman, Florham Park, NJ, USA) before performing the HPLC analysis. Both mycotoxin standards and diluted media samples were analyzed using an Agilent HPLC system (1200 Series, Palo Alto, CA, USA) equipped with a quaternary pump, an inline degasser, and a Diode Array Detector (DAD) with a wavelength set at 218 nm. Both a Phenomenex^®^ 4µ Jupiter Proteo 90A (250 × 4.6 mm) column equipped with a C18-guard column (Phenomenex, Torrance, CA, USA) were used for the separation. Compounds of interest were eluted using a binary mobile-phase composed of acetonitrile:water (1:9 by volume for DON and 3-*epi*-DON, respectively, and 2:8 by volume for 3-keto-DON). The flow rate was fixed to 1.0 mL/min while the sample injection volume was set at 10 μL.

### 4.5. Lemna minor Transcriptome Sequencing and Analysis

*Lemna**minor* plants were cultured as described above for 7 days to prepare the inoculum. On the 8th day, 2 g of the plant’s vegetative-material was transferred into flasks containing 100 mL of freshly made Hoagland’s E+ medium supplemented with either 2.5 µg/mL DON, 2.5 µg/mL 3-keto-DON, or 2.5 µg/mL 3-*epi*-DON, respectively. The negative control contained Hoagland’s E+ medium alone. After 24 h, 500 mg of plant tissues were collected from each treatment (3 replicates from 3 separate experiments) and were snap frozen in liquid nitrogen at −80 °C.

Frozen *L. minor* samples were shipped on dry-ice to Novogene Corporation Inc (Sacramento, CA, USA) for RNA-Seq analysis. Total RNA extraction, library preparation, and sequencing were all performed according to the service-provider’s standard protocols. Briefly, total RNA (1 μg) was prepared and used to generate sequencing-libraries using the NEBNext Ultra RNA Library Prep Kit from (Ipswich, MA, USA) following the manufacturer’s instructions. The library quality was assessed on an Agilent Bioanalyzer 2100 system. Clustering was performed on a cBot Cluster Generation System using PE Cluster Kit cBot-HS kit (Illumina) according to the manufacturer’s instructions. After cluster generation, library preparations were sequenced on an Illumina platform to generate 150 bp paired-end reads. The resulting reads were quality filtered to remove adaptor sequences and any read pairs containing >10% Ns or where ≥50% of bases had a ≤Q20 score. Filtered reads were mapped to the *L. minor* genome (https://genomevolution.org/coge/ accessed on 22 October 2019; ID: 27408) with HISAT2 (version 2.1.0) [56], and HTSeq (version 0.6.1) [57] was used to determine FPKM values. Differentially expressed genes (DEGs) between the control and treatment groups were identified using the DESeq R package (version 1.10.1) and *p*-values were adjusted by the Benjamini–Hochberg (BH) method [58] where genes with a *p*_adj_ < 0.001 and Abs(fold-change) > 5 were considered significant. Multidimensional scaling (MDS) of the gene expression results was performed with Degust [59]. Gene ontology (GO) and KEGG pathway over-representation analysis (ORA) was performed with the clusterProfiler R package (version 3.18) [60], where GO terms and KEGG pathways with a BH-corrected *p*-value < 0.05 were considered significantly enriched. The annotated KEGG pathways were generated with the Pathview R package (version 1.30) [61].

### 4.6. Regression and Statistical Analysis

The inhibitory effects of DON, 3-keto-DON, and 3-*epi*-DON towards the growth of *L. minor* fronds were evaluated by a detailed regression analysis using SPSS (Version 27.0, IBM Corp.). The exponential model was found to fit well within the collected response-points of the three aforementioned compounds. The obtained model formula was represented by:ln(Y)=A×X+ln(B)
where *X* is the concentration of the mycotoxins (µg/mL), *Y* is the relative growth index (RGI), which is defined to be: the ratio between the number of grown-fronds to negative-control samples, *A* is the coefficient, and *B* is the constant. The differences between IC_50_ (50% inhibition of *L*. *minor* fronds growth) means were evaluated using the SPSS package through an independent-sample *t*-test. The phytotoxic effects of DON and its metabolites on the *L. minor* and wheats were evaluated through one-way ANOVA with post-hoc Tukey’s HSD test.

## 5. Conclusions

The current work provides a unique prospective on the toxicology of 3-keto-DON and 3-*epi*-DON, two bacterial metabolites of DON. While earlier studies have showed a substantial reduction of DON cellular toxicity due to the double bond formation of the C3 carbon (3-keto-DON) or its isomerization (3-*epi*-DON), our findings argue about the importance of the host-organism and the substantial role its molecular pathways play. A future scrutiny of the influencing mechanism(s) and the effect of the above bacterial metabolites on plant productivity/yield (wheat/corn) are granted.

## Figures and Tables

**Figure 1 ijms-23-07230-f001:**
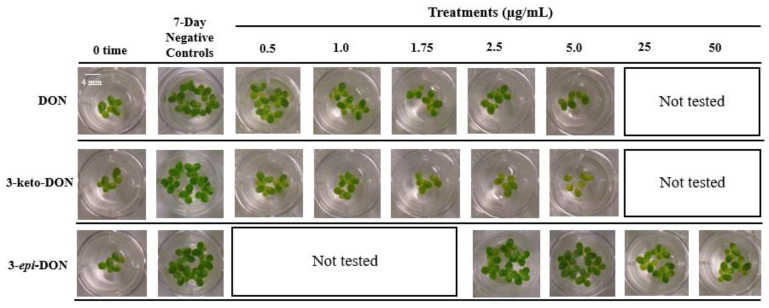
A comparative study of the phytotoxic effects of DON, 3-keto-DON, and 3-*epi*-DON on the growth and morphology of *Lemna minor*. The response of duckweed towards DON and its biotransformation products (3-keto-DON and 3-*epi*-DON) was tracked within 7 days of exposure. Scale bar = 4 mm.

**Figure 2 ijms-23-07230-f002:**
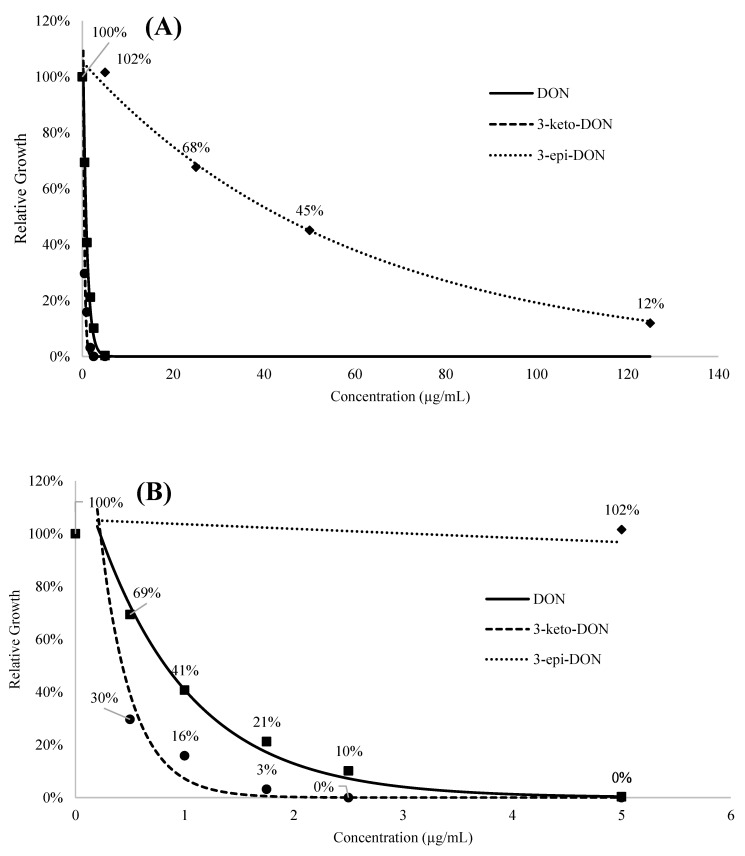
Regression curves of relative growth indices (RGIs) under the exposure of DON, 3-keto-DON, and 3-*epi*-DON are shown: (**A**) the full range of regression curves of obtained RGIs under the reported experimental conditions; (**B**) showing RGIs for the range between 0 and 5 µg/mL of the three tested compounds.

**Figure 3 ijms-23-07230-f003:**
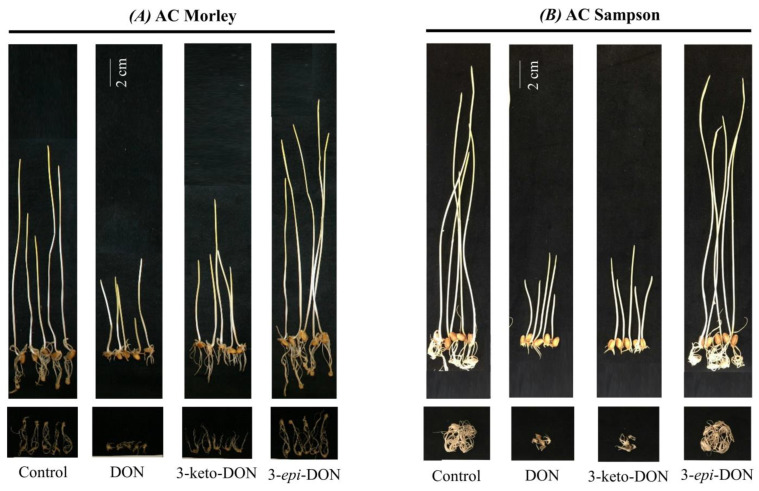
The phytotoxic effects of DON and its biotransformation products, 3-keto-DON and 3-*epi*-DON, on the growth and morphology of wheat seedlings. The two wheat cultivars that were tested are AC Morley (**A**) (Winter wheat, moderately resistant to *Fusarium*) and AC Sampson (**B**) (Winter wheat, susceptible to *Fusarium*). Each seedling (*n* = 15) was grown from germinated seeds for seven days on gellan gum-soft gel medium (GSGM) containing DON, 3-keto-DON, or 3-*epi*-DON and 1% DMSO at 28 °C. The shown photographs are representatives of the seedlings from one group of experiments, scale bar = 2 cm.

**Figure 4 ijms-23-07230-f004:**
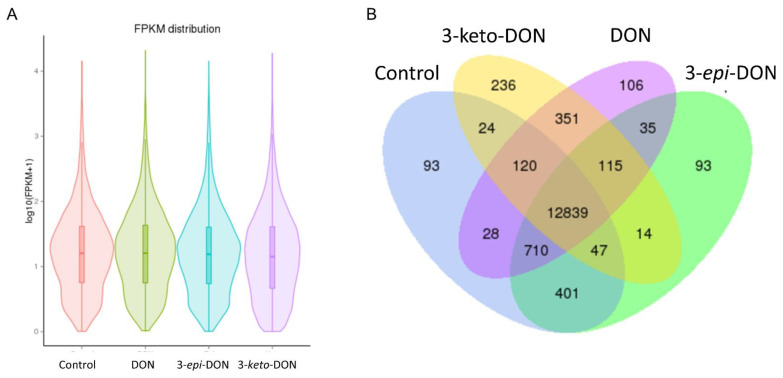
Distribution of expressed genes among *L. minor* plants exposed to DON, 3-keto-DON, 3-*epi*-DON, or media alone: (**A**) violin plot of mean log2 FPKM values between treatment groups; (**B**) Venn diagram indicating the number of genes co-expressed between different treatment groups.

**Figure 5 ijms-23-07230-f005:**
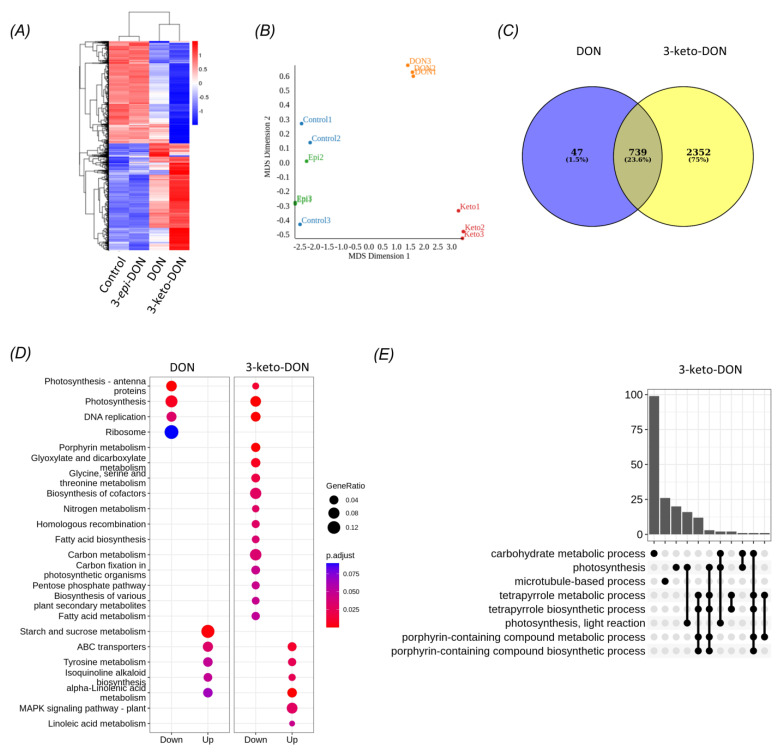
Significantly differentially-expressed genes among *L. minor* plants exposed to DON, 3-keto-DON, 3-*epi*-DON, or media alone: (**A**) Clustered heatmap of expression values (log10(FPKM + 1)), where red and blue represent high and low expression levels, respectively; (**B**) multidimensional scaling (MDS) plot of gene expression results representing variation between samples; (**C**) Venn diagram indicating the number of DEGs in common between DON and 3-keto-DON groups compared to control; (**D**) significantly enriched (BH-adjusted *p*-value ≤ 0.05) KEGG pathways associated with DEGs from DON and 3-keto-DON groups compared with control. The size of the circle indicates the proportion of DEGs associated with the pathway compared to the total geneset, and the color indicates adjusted *p*-value (**E**) UpSet plot of significantly enriched (adjusted *p*-value ≤ 0.05) GO biological process terms associated with DEGs from 3-keto-DON groups compared with control. Bars indicate the total genes shared between GO processes, specified by connected dots.

**Figure 6 ijms-23-07230-f006:**
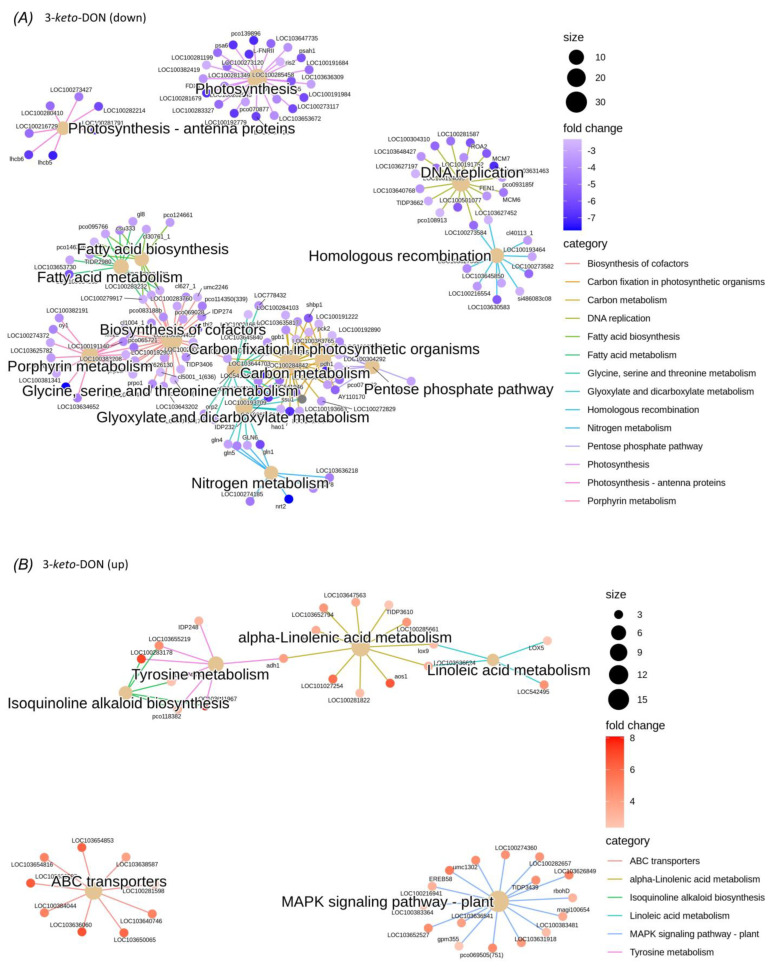
Gene concept networks of significantly enriched KEGG pathways among *L. minor* plants exposed to 3-keto-DON. Networks illustrate (**A**) down-regulated or (**B**) up-regulated DEGs associated with enriched KEGG pathways 3-keto-DON-treated group. Central nodes indicate KEGG pathways, with size representing the total number of associated DEGs, and outer nodes indicate associated DEGs, with colors representing fold-change compared to control group.

**Table 1 ijms-23-07230-t001:** The effect of DON and 3-keto-DON on *Lemna minor* L. growth after one week of co-incubation.

Experiment		Fronds per Well (Mean ± SD)	Fresh-Weight in mg/Well (Mean ± SD)
**(A)** **DON concentration (µg/mL)**	0	24.56 ± 4.45 ^a^	18.33 ± 4.38 ^a^
	0.5	18.88 ± 3.07 ^ab^	13.75 ± 2.23 ^ab^
	1	13.56 ± 2.54 ^bcd^	9.50 ± 1.47 ^bc^
	1.75	9.94 ± 1.73 ^cde^	6.22 ± 1.03 ^c^
	2.5	7.88 ± 1.11 ^cde^	5.64 ± 0.82 ^c^
	5	6.06 ± 0.13 ^de^	4.46 ± 0.70 ^c^
**(B)** **3-keto-DON concentration (µg/mL)**	0	21.75 ± 3.03 ^a^	14.57 ± 2.74 ^a^
	0.5	10.67 ± 1.70 ^b^	7.19 ± 0.46 ^b^
	1	8.50 ± 1.30 ^bc^	5.95 ± 1.01 ^b^
	1.75	6.50 ± 0.66 ^bc^	4.84 ± 0.84 ^b^
	2.5	6.00 ± 0.00 ^c^	4.43 ±0.49 ^b^
	5	6.00 ± 0.00 ^c^	3.78 ± 0.95 ^b^

All data are the mean ± SD, *n* = 3. Different letters (^a–e^) represent significant differences in the same experiment within rows (*p* < 0.05). Values that share the same letter are not significantly different.

**Table 2 ijms-23-07230-t002:** Growth and the final fresh-weight of *Lemna minor* L. plants exposed to 3-*epi*-DON for seven days.

3-*epi*-DON Concentration (µg/mL)	Fronds per Well	Percentage of Control (%)	Fresh Weight in mg/Well	Percentage of Control (%)
**0**	24.83	100.0 (control)	18.26	100.0 (control)
**5**	25.13	101.2	18.05	98.8
**25**	18.75	75.5	12.37	67.7
**50**	14.5	58.4	8.5	46.6
**125**	8.25	33.2	5.38	29.5

**Table 3 ijms-23-07230-t003:** The effect of DON and metabolites, 3-keto-DON and 3-*epi*-DON, on coleoptile length (mm/seedling), coleoptile’s fresh-weight (mg/seedling), and root’s fresh-weight (mg/seedling).

**AC Morley (Moderately-Resistant)**			
**Treatment/Measurements** **[mean ± SD]**	**Coleoptile’s Length (mm/seedling)**	**Coleoptile’s Fresh-Weight mg/seedling)**	**Root’s Fresh-Weight** **(mg/seedling)**
**Control**	130.75 ± 8.83 ^a^	73.54 ± 1.70 ^a^	36.60 ± 8.80 ^a^
**DON (30 µg/mL)**	60.73 ± 12.21 ^b^	48.29 ± 8.24 ^b^	17.49 ± 6.19 ^b^
**3-keto-DON (30 µg/mL)**	68.57 ± 13.05 ^b^	55.95 ± 2.20 ^b^	21.90 ± 5.11 ^a^
**3-*epi*-DON (30 µg/mL)**	130.21 ± 20.40 ^a^	70.92 ± 5.75 ^a^	33.92 ± 5.52 ^a^
**AC Sampson (Susceptible)**			
**Treatment/Measurements** **[mean ± SD]**	**Coleoptile’s Length (mm/seedling)**	**Coleoptile’s Fresh-Weight mg/seedling)**	**Root’s Fresh-Weight (mg/seedling)**
**Control**	157.90 ± 3.92 ^a^	69.84 ± 5.37 ^a^	33.00 ± 4.60 ^a^
**DON (30 µg/mL)**	48.97 ± 3.95 ^b^	39.28 ± 1.44 ^b^	5.77 ± 0.50 ^b^
**3-keto-DON (30 µg/mL)**	52.47 ± 3.00 ^b^	38.53 ± 4.93 ^b^	5.13 ± 1.16 ^b^
**3-*epi*-DON (30 µg/mL)**	165.48 ± 5.41 ^a^	76.57 ± 4.96 ^a^	41.62 ± 9.45 ^a^

All data are the mean ± SD, *n* = 3. Different letters within rows represent significant differences within the same measured parameters (*p* < 0.05). Values that share the same letter are not significantly different.

## Data Availability

Not applicable.

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
