# Peer review of "3-keto-DON, but Not 3-epi-DON, Retains the in Planta Toxicological Potential after the Enzymatic Biotransformation of Deoxynivalenol"

_ijms, 2022, doi:10.3390/ijms23137230_

Round 1

Reviewer 1 Report

A very important group of sesquiterpene derivatives are trichothecenes. Due to the toxicity of these metabolites, their presence in cereal products and derivatives poses a significant threat / problem and is the reason for introducing regulatory limits for trichothecene mycotoxins, mainly DON. Any solutions that lead to detoxification should be considered the most desirable and necessary. The presented manuscript brings new knowledge in this area. It focuses on the toxicity of DON derivatives (3-keto-DON and 3-epi-DON) in a two-model system, and the presented results fully deserve publication

Author Response

We thank the knowledgeable reviewer for his comments. We totally agree with his point regarding the importance of addressing the toxicity of trichothecenes (and derivatives) in cereals products as they pose a significant threat/problem for the agricultural field/cereals trading in general.

Reviewer 2 Report

An interesting and relevant report by Li et al., on the impact of select naturally occurring mycotoxin derivatives (biodetoxification products) highlighting new bioassays, differential environmental toxicities and mechanisms underlying these effects. Novelty lies with the environmental impact on plants, compared to most prior studies that have focussed on toxicity in animals. The report is very well written. Excellent controls and statistics throughout.  A few minor comments below.

Table 1 – The line defining statistical annotations is not entirely clear. Is it that values that share the same letter are NOT significantly different? Clarify. Same issue for Table 3.

Figure 5 – I really appreciated the succinct description of the RNA-Seq results, leaving a lot of the detail (gene names etc) out of the text, enabling me to readily see the bigger picture.  However Figure 5 is so small – even when viewing the PDF document at 150% on my laptop that I can hardly read the gene name/GO details in the figure, leaving me a bit frustrated about exactly what pathways and genes have been modified.  Perhaps Figure 5 needs to be broken up into multiple figures – with the individual images shown much bigger, or some of the images moved to the supplemental and shown in a bigger format there?

I appreciated the review of current literature regarding RNAseq analyses of DON treated plants in the discussion.

Methods are good. No concerns here.

Figures A1 and A2 are indecipherable in the PDF format due to being WAY too small and when I increase to 400% it is pixelated and still indecipherable. Again – better to break these figures down into individual figures that are bigger and readable in the PDF format. Alot of effort went into making these figures.  Let us see them!

Line 24 :  revise to read ‘…plant applications. Collectively they…’

Lin 221: revise to read ‘…This was also apparent in…’

Line 302: revise to read ‘…A previous…’

Author Response

Table (1)- The line defining statistical annotations is not entirely clear. Is it that values that share the same letter are not significantly different? Clarify. Same issue for Table 3:

Response: We added the following sentence “values that share the same letter are not significantly different” to Table 1 and Table 3, respectively.

Figure 5- I really appreciated the succinct description of the RNA-Seq results, leaving a lot of the detail (gene names etc) out of the text, enabling me to readily see the bigger picture.  However, Figure 5 is so small-even when viewing the PDF document at 150% on my laptop that I can hardly read the gene name/GO details in the figure, leaving me a bit frustrated about exactly what pathways and genes have been modified.  Perhaps Figure 5 needs to be broken up into multiple figures-with the individual images shown much bigger, or some of the images moved to the supplemental and shown in a bigger format there?

Response: We introduced the necessary changes requested by the respected reviewer to the aforementioned figure in order to make it easier to read/follow. The original figure #5 was split into two separate figures (now #5 and #6, respectively).  The images in Figures 5 D/E and 6 A/B were altered to increase font size.

Figures A1 and A2 are indecipherable in the PDF format due to being way too small and when I increase to 400% it is pixelated and still indecipherable. Again, better to break these figures down into individual figures that are bigger and readable in the PDF format. A lot of effort went into making these figures.  Let us see them:

Response: We introduced changes to the above figures, as requested by the respected reviewer, to make them more legible. An additional supplemental figure was also added (A3) to show gene concept networks of significantly enriched KEGG pathways among L. minor plants exposed to DON.

Line 24:  revise to read ‘…plant applications. Collectively they…’

Response: Addressed.

Lin 221: revise to read ‘…This was also apparent in…’

Response: Addressed.

Line 302: revise to read ‘…A previous…’

Response: Addressed.

Reviewer 3 Report

Please, be aware of some minor spelling and grammar error at the beginning of the manuscript.  Additionally, some figures do not read well, specially Figure 5 and Figures A1 and A2, because they are too crowded or some images are too small for the reader. It might be necessary to split them in two or three parts/sections.

Author Response

Please, be aware of some minor spelling and grammar error at the beginning of the manuscript.

Response: We have proofread the manuscript and went over the entire text to eliminate any typos/grammatical errors. Changes are highlighted in yellow.

Additionally, some figures do not read well, specially Figure 5 and Figures A1 and A2, because they are too crowded, or some images are too small for the reader. It might be necessary to split them in two or three parts/sections.

Response: We introduced changes to the aforementioned figures, as requested by the respected reviewer, in order to make them easier to read/follow:

The original Figure #5 was split into two separate figures (now #5 and #6, respectively).  The images in Figures 5 D/E and 6 A/B were altered to increase font size.

We introduced changes to Figures A1 and A2, to make them more legible. An additional supplemental figure was also added (A3) to show gene concept networks of significantly enriched KEGG pathways among L. minor plants exposed to DON.

This manuscript is a resubmission of an earlier submission. The following is a list of the peer review reports and author responses from that submission.

Round 1

Reviewer 1 Report

This is very good and valuable paper and significantly expands the knowledge on the detoxification of trichothecenes. Each part of the manuscript (especially results and discussion) were carefully and very clearly elaborated. The results are fully explained and comprehensive and correct interpreted . In conclusion with full responsibilities  I can recommend this paper for publication.

my minor remarks:

Introduction provide information that DON is Fusarium toxin. It is not true for all Fusaria, not  all species of this genus  are DON producers. Please consider modification of this shortcuts.

Some key words are repetition of words used in title (3-epi-DON; 3-keto- DON) perhaps it is worth to use another one.

Author Response

We greatly appreciate the reviewer's time, efforts and encouragement, and have revised the manuscript as indicated below:

This is very good and valuable paper and significantly expands the knowledge on the detoxification of trichothecenes. Each part of the manuscript (especially results and discussion) was carefully and very clearly elaborated. The results are fully explained and comprehensively  and correctly interpreted. In conclusion with full responsibilities, I can recommend this paper for publication.

My minor remarks:

*** Introduction: Provide information that DON is Fusarium toxin. It is not true for all Fusaria, not all species of this genus are DON producers. Please consider modification of this shortcuts:

The phrase “a Fusarium toxin produced by F. culmorum and F. graminearum” was introduced based on the reviewer’s recommendation.

*** Some key words are repetition of words used in title (3-epi-DON; 3-keto- DON) perhaps it is worth to use another one:

The note was taken and a modification (the use of stereoisomer instead of 3-epi-DON) was introduced.

Reviewer 2 Report

This manuscript compared the toxicity of DON detoxification metabolites (3-keto-DON, 3-epi-DON) on plant models (Lemna minor and wheat). Very interesting work on DON toxicity research. DON is a very stable mycotoxin in cereal-based food and feed, and the oxidation and epimerization pathways are well-known detoxification processes on DON. In this work, authors have done comprehensive work concerning DON toxicity effects on plants, which could be accepted, but need some improvement.

In the section of the Introduction, please provide more background about DON biodetoxification, including the reported DON detoxification microbes and related pathways. You also can make a schematic figure on popular DON detoxification processes.

In figure 1, I see the morphology of Lemna minor. 3-keto-DON seems more toxic than DON? Please check it.

Line 154, why you only use the concentration of 30 μg/mL? Did you try other concentrations? Such as 10 or 20 μg/mL.

Line 291 to 295, it is better to cite more papers on cell model-based toxicity evaluation of 3-keto-DON and 3-epi-DON.

Figure 5. This is a summarized figure of DEGs via Transcriptome Sequencing and Analysis, I didn’t see the PCA(principal component analysis) which can directly present the profile changes of control and treated groups, please provide it.

In the section of Discussion, we know PQQ is an essential cofactor on converting DON to 3-keto-DON, and in the second step, NADPH is needed on 3-keto-DON to 3-epi-DON. It is challenging in real application environment because we need to add these additional ingredients for DON detoxification. Secondary, though 3-epi-DON is less toxic, the efficiency of two-steps on converting DON to 3-epi-DON is unwarrantable. So please discuss the difficulty and acceptable solutions for enzymes application to on detoxification DON of into 3-epi-DON.

Author Response

We greatly appreciate the reviewer’s time, efforts and suggestions, and have revised the manuscript as indicated below:

This manuscript compared the toxicity of DON detoxification metabolites (3-keto-DON, 3-epi-DON) on plant models (Lemna minor and wheat). Very interesting work on DON toxicity research. DON is a very stable mycotoxin in cereal-based food and feed, and the oxidation and epimerization pathways are well-known detoxification processes on DON. In this work, authors have done comprehensive work concerning DON toxicity effects on plants, which could be accepted, but need some improvement:

 *** In the Introduction, please provide more background about DON biodetoxification, including the reported DON detoxification microbes and related pathways. You also can make a schematic figure on popular DON detoxification processes.

The reviewer has a good point over here and considering his comment, we have updated the introduction to reasonably enhance its content (UDP-glycosyltransferases for example) in the light of the above recommendation. Unfortunately, due to the many figures and diagrams that we already have within this manuscript, we could not include any additional figures it would form rather a distraction over here.

*** In figure 1, I see the morphology of Lemna minor. 3-keto-DON seems more toxic than DON? Please check it:

We agree with the reviewer’s observation. The repeated experiments always indicated that 3-keto-DON is in fact slightly more toxic than the parental compound, DON under the reported experimental conditions.

 *** Line 154, why you only use the concentration of 30 μg/mL? Did you try other concentrations? Such as 10 or 20 μg/mL?:

Yes, different concentrations were tried including 10 μg/mL, 20 μg/mL, 30 μg/mL and 100 μg/mL; only 30 μg/mL results showed here.

 *** Line 291 to 295, it is better to cite more papers on cell model-based toxicity evaluation of 3-keto-DON and 3-epi-DON:

 Two additional references and a clarifying sentence were added based on the reviewer’s recommendation (lines 305-306).

 *** Figure 5. This is a summarized figure of DEGs via Transcriptome Sequencing and Analysis, I didn’t see the PCA(principal component analysis) which can directly present the profile changes of control and treated groups, please provide it:

As suggested by the reviewer, a multidimensional scaling (MDS) plot, which serves an equivalent purpose to PCoA, has been added and labelled as Figure 5B

 *** In the section of Discussion, we know PQQ is an essential cofactor on converting DON to 3-keto-DON, and in the second step, NADPH is needed on 3-keto-DON to 3-epi-DON. It is challenging in real application environment because we need to add these additional ingredients for DON detoxification. Secondary, though 3-epi-DON is less toxic, the efficiency of two-steps on converting DON to 3-epi-DON is unwarrantable. So please discuss the difficulty and acceptable solutions for enzymes application to on detoxification DON of into 3-epi-DON:

A paragraph acknowledging the above concerns was added (lines 384-388).

Round 2

Reviewer 2 Report

 *** Line 154, why you only use the concentration of 30 μg/mL? Did you try other concentrations? Such as 10 or 20 μg/mL?:

Yes, different concentrations were tried including 10 μg/mL, 20 μg/mL, 30 μg/mL and 100 μg/mL; only 30 μg/mL results showed here.

Response: How about the results of other DON concentrations of treatment?

 *** Line 291 to 295, it is better to cite more papers on cell model-based toxicity evaluation of 3-keto-DON and 3-epi-DON:

 Two additional references and a clarifying sentence were added based on the reviewer’s recommendation (lines 305-306).

Response: Citations need more description in the section of Discussion.

*** In the section of Discussion, we know PQQ is an essential cofactor on converting DON to 3-keto-DON, and in the second step, NADPH is needed on 3-keto-DON to 3-epi-DON. It is challenging in real application environment because we need to add these additional ingredients for DON detoxification. Secondary, though 3-epi-DON is less toxic, the efficiency of two-steps on converting DON to 3-epi-DON is unwarrantable. So please discuss the difficulty and acceptable solutions for enzymes application to on detoxification DON of into 3-epi-DON:

A paragraph acknowledging the above concerns was added (lines 384-388).

Response: Lines 384-388, I am afraid that this is insufficient to address my questions before being formally accepted. The authors only raised the concerns of cofactors in application that I mentioned but didn’t discuss the efficiency and difficulty of two-steps detoxifying DON to 3-epi-DON and possible solutions for adding cofactors in the detoxification process. Authors in this work stated that 3-keto-DON is toxic than 3-epi-DON, these issues should be comprehensively discussed.

Author Response

Response to reviewer’s additional comments (in blue)
*** Line 154, why you only use the concentration of 30 μg/mL? Did you try other concentrations? Such as 10 or 20 μg/mL?:
Yes, different concentrations were tried including 10 μg/mL, 20 μg/mL, 30 μg/mL and 100 μg/mL; only 30 μg/mL results showed here.
Response: How about the results of other DON concentrations of treatment?
The 10 and 20 ug/mL DON concentrations showed same tendency as we have witnessed in the 30 ug/mL.
As this paper already contains large amount of data, we feel it is not necessary to include all results. A statement is added to the results (line 162 - 164)
*** Line 291 to 295, it is better to cite more papers on cell model-based toxicity evaluation of 3-ketoDON and 3-epi-DON:
Two additional references and a clarifying sentence were added based on the reviewer’s recommendation (lines 305-306).
Response: Citations need more description in the section of Discussion.
Descriptions related to the two reference were added (line 307-309)
*** In the section of Discussion, we know PQQ is an essential cofactor on converting DON to 3-ketoDON, and in the second step, NADPH is needed on 3-keto-DON to 3-epi-DON. It is challenging in real application environment because we need to add these additional ingredients for DON detoxification.
Secondary, though 3-epi-DON is less toxic, the efficiency of two-steps on converting DON to 3-epi-DON is unwarrantable. So please discuss the difficulty and acceptable solutions for enzymes application to on detoxification DON of into 3-epi-DON:
A paragraph acknowledging the above concerns was added (lines 384-388).
Response: Lines 384-388, I am afraid that this is insufficient to address my questions before being formally accepted. The authors only raised the concerns of cofactors in application that I mentioned but didn’t discuss the efficiency and difficulty of two-steps detoxifying DON to 3-epi-DON and possible solutions for adding cofactors in the detoxification process. Authors in this work stated that 3-keto-DON is toxic than 3-epi-DON, these issues should be comprehensively discussed.
We do appreciate the reviewer’s questions and have added more discussion (386 -395).
